

# Gene flow between subpopulations of gray snapper (*Lutjanus griseus*) from the Caribbean and Gulf of Mexico

Oscar de Jesús Rosado-Nic[1], J. Derek Hogan[2], José Héctor Lara-Arenas[1], Rigoberto Rosas-Luis[1,3], Laura Carrillo[4] and Carmen Amelia Villegas-Sánchez[1]

[1] Tecnológico Nacional de México/IT de Chetumal, Chetumal, Quintana Roo, Mexico
[2] Department of Life Sciences, Texas A&M University–Corpus Christi, Corpus Christi, TX, USA
[3] Cátedras Conacyt–Tecnológico Nacional de México/IT de Chetumal, Chetumal, Quintana Roo, Mexico
[4] Departamento de Sistemática y Ecología Acuática, El Colegio de la Frontera Sur, Chetumal, Quintana Roo, Mexico

## ABSTRACT

**Background:** The gray snapper (*Lutjanus griseus*) has a tropical and subtropical distribution. In much of its range this species represents one of the most important fishery resources because of its high quality meat and market value. Due to this, this species is vulnerable to overfishing, and population declines have been observed in parts of its range. In recent decades, it has been established that knowing the level of genetic connectivity is useful for establishing appropriate management and conservation strategies given that genetic isolation can drive towards genetic loss. Presently the level of genetic connectivity between subpopulations of *L. griseus* of the southern region of the Gulf of Mexico and the Caribbean Sea remains unknown.
**Methods:** In the present study we analyze genetic structure and diversity for seven subpopulations in the southern Gulf of Mexico and the Mexican Caribbean Sea. Eight microsatellite primers of phylogenetically closely related species to *L. griseus* were selected.
**Results:** Total heterozygosity was 0.628 and 0.647 in the southern Gulf of Mexico and the Mexican Caribbean Sea, however, results obtained from AMOVA and $R_{ST}$ indicated a lack of genetic difference between the major basins. We also found no association between genetic difference and geographic distance, and moderately high migration rates ($N_m = > 4.1$) suggesting ongoing gene flow among the subpopulations. Gene flow within the southern Gulf of Mexico appears to be stronger going from east-to-west.
**Conclusions:** Migration rates tended to be higher between subpopulations within the same basin compared to those across basins indicating some regionalization. High levels of genetic diversity and genetic flow suggest that the population is quite large; apparently, the fishing pressure has not caused a bottleneck effect.

Corresponding author
Carmen Amelia Villegas-Sánchez,
cvillegas@itchetumal.edu.mx

## INTRODUCTION

Establishing effective fisheries regulations is a complex and multidisciplinary task (*INAPESCA, 2012*). In recent decades, it has been determined that delineating stock boundaries and knowing the level of genetic connectivity among stocks is useful for establishing appropriate management and conservation strategies (*Villegas Sánchez et al., 2014*). Improper management can lead to genetic diversity loss and increased inbreeding within genetically isolated populations with negative effects for the survival of the populations (*Urbiola-Rangel & Chassin-Noria, 2013*; *Villegas Sánchez et al., 2014*). With the use of genetic markers, diversity and the level of genetic connectivity can be estimated between populations at different geographic scales.

The Gray snapper (*Lutjanus griseus*) is an economic and ecologically important fishery species and can be highly abundant throughout its range, yet there is a paucity of information for its fishery management (*Lindeman et al., 2016*). Gray snapper fisheries are subjected to fishery regulations in some countries, for example, in the Everglades National Park, United States, a catch limit of 10 individuals per person was established in the 1970's (*Claro & Lindeman, 2008*). In Cuba the fishery is closed in June, during the breeding season (*Claro & Lindeman, 2008*). Nevertheless, in Mexico, although it's being captured by fishers, it has not been classified as overexploited or subjected to fishery overexploitation, thus no closed season or other regulation has been established (*INAPESCA, 2018*); this lack of regulation could place populations at risk in the future (*Costello et al., 2012*).

The gray snapper presents a wide distribution from North Carolina, United States to southern Brazil, in the countries where it is found it represents an important reef fishery resource because of its high quality meat and market value (*Claro & Lindeman, 2008*). The gray snapper is a predator that feeds on a wide variety of organisms in different habitats including estuaries, mangroves and seagrass beds, whereof changes in their populations have great impacts in other elements of the community (*Claro & Lindeman, 2008*; *Rocha & Molina, 2008*). Migratory movements of adult individuals are mainly local and sexual maturity occurs when they are 1 year old reaching a total length between 260 and 280 mm. The pelagic larval period of this species is 25–26 days and larval settlement has been recorded between 30 and 40 days after hatching (*Claro & Lindeman, 2008*). Its large geographic range and month-long larval duration indicate that connectivity among populations may be widespread.

Despite its importance as a fisheries species, there is relatively little knowledge about the stock structure of gray snapper, however stocks appear to be declining in some places. Commercial landings in the US South Atlantic region (North Carolina to Florida) have been declining since the 1950's and populations in the Florida Keys are thought to be potentially over-fished (*Ault, Bohnsack & Meester, 1998*). In Cuba, gray snapper is abundant; however, fisheries landings have declined over the past 30 years as have spawning aggregations (*Claro et al., 2009*). In Mexico, there is currently no information about stock structure and gray snappers are caught in a mixed stock fishery (*FAO, 2011*; *SAGARPA, 2012*), the health of which is not well known. There are few population genetics studies for the gray snapper and only one study related to genetic connectivity

exists (*Gold et al., 2009*). The authors looked at genetic structure at a scale of 2,400 km in the north of the Gulf of Mexico and the U.S. Atlantic using microsatellites, reporting the existence of genetically different populations and a decrease in their effective size from east to west presumably driven by ocean currents. They recommended that these populations be treated as independent stocks for effective management of the fishery. However, there have been no studies of genetic structure or connectivity of gray snapper in the southern Gulf of Mexico or Mexican Caribbean.

In Mexico, we hypothesize that there may be genetic differences between the populations of gray snapper in the Mexican Caribbean and the southern Gulf of Mexico. Prior studies have shown that there is genetic differentiation between these major basins in other taxa. Blacktip sharks (*Carcharhinus limbatus*), a low dispersal species, show strong genetic differentiation between the Mesoamerican Barrier Reef System and the southern Gulf of Mexico (*Keeney et al., 2005*). The bicolor damselfish (*Stegastes partitus*), a high dispersal reef fish (*Hogan et al., 2012*), has shown evidence of a weak restriction in gene flow between the Mexican Caribbean and southern Gulf of Mexico (*Villegas Sánchez et al., 2014*). Similarly, the lionfish (*Pterois volitans*), the most studied invasive species, has been reported as having significant genetic differentiation between both regions, which suggests a phylogeographic break (*Labastida-Estrada et al., 2019*). The objective of this study is to determine the diversity and genetic connectivity among seven subpopulations of gray snapper in the southern Gulf of Mexico and Mexican Caribbean Sea.

# MATERIALS AND METHODS

## Study area

Two regions were studied: (1) Gulf of Mexico, with the subpopulations Campeche (*C*), Puerto de Veracruz (*PV*) and Tuxpan (*TX*); (2) Mexican Caribbean Sea, with the subpopulations Bahia de Chetumal (*BC*), Xahuayxol (*X*), Punta Herrero (*PH*) and Chiquilá (*CH*) (Fig. 1). Our seven sampling sites are distributed along approximately 1,950 km of coast between the Caribbean Sea and southern Gulf of Mexico. The Gulf of Mexico region possesses an extensive continental shelf with a diversity of ecosystems like wetlands, the largest area of mangroves in Mexico, coastal dunes and coral reefs with a tropical and subtropical climate (*Lara-Lara et al., 2008*). The Mexican Caribbean region with a narrow continental shelf is located in the Southeast of Mexico, with a warm subhumid climate and an annual average temperature of 26 °C and a mean annual precipitation of 1,300 mm (*Lara-Lara et al., 2008*).

## Fish sampling and sample storage

Muscular tissue was taken from the base of the caudal fin of 348 organisms captured by fishermen (50 samples per site, except for the *X* site, with 48 samples; SEMARNAT approved the field study with the number: 23/K4-0002/05/18). The sampling was carried out during November and December 2016 for the Mexican Caribbean Sea region, and during the same months 2017 for the Southern Gulf of Mexico region. Such periods were chosen in order to only catch resident individuals so as not to bias the estimates of
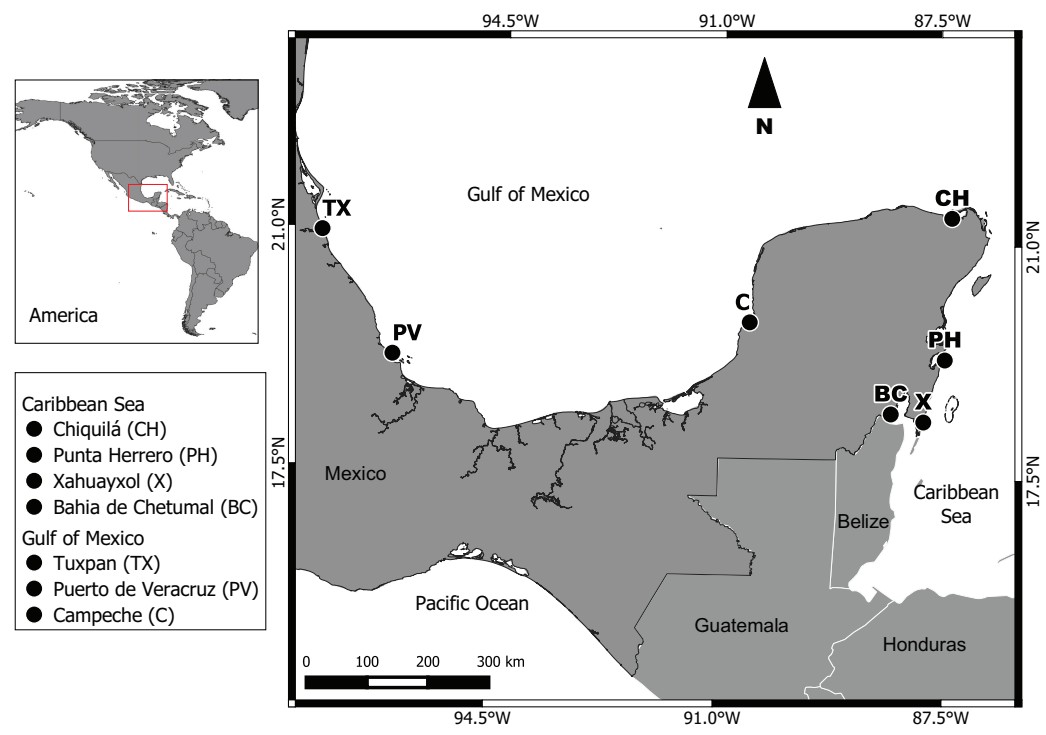

**Figure 1  Geographic location of the seven sampling sites.**

gene flow, given that the migration of the gray snapper occurs from May to October (*Claro & Lindeman, 2008*; *Espinoza Ávalos, 2009*). All samples were preserved in 95% ethanol and stored at −5 °C for transportation and laboratory storage prior to DNA extraction.

## DNA isolation

DNA was isolated with the Qiagen DNeasy Blood & Tissue Kit following manufacturer's protocols. DNA concentration and quality was verified using a spectrophotometer (Thermo Scientific NanoDrop™ 2000, Waltham, MA, USA).

## Molecular markers

Microsatellites are commonly used DNA markers in genetic diversity studies because they present greater number of polymorphs (allelic variation) per locus than other markers. This feature makes them more sensitive to changes in size, structure and dispersion rate of populations (*Goldstein & Schlötterer, 1999*). Additionally, microsatellites are co-dominant markers which allow to distinguish between homozygous and heterozygous individuals (*Estoup et al., 1998*; *Goldstein & Schlötterer, 1999*).

Eight microsatellite primers were selected from published literature of phylogenetically closely related species to *L. griseus*, which were selected for their high levels of polymorphism and their ability to amplify in *L. griseus*. Seven loci were developed from the red snapper (*Lutjanus campechanus*) while one locus (Ra1) was developed from the

**Table 1 Genetic diversity of the gray snapper (*Lutjanus griseus*) in the southern Gulf of Mexico.**

| Sites | | Lca20 58 °C | Lca43 56 °C | Prs260 56 °C | Ra1 58 °C | Lca107 48 °C | Prs137 54 °C | Prs275 54 °C | Prs328 54 °C | Mean |
|---|---|---|---|---|---|---|---|---|---|---|
| C | $Na$ | 5 | 2 | 7 | 9 | 8 | 9 | 4 | 3 | 5.875 |
| | $A_E$ | 3.814 | 1.259 | 5.300 | 2.312 | 2.970 | 5.545 | 2.784 | 2.220 | 3.275 |
| | $Ho$ | 0.720 | 0.186 | 0.837 | 0.542 | 0.327 | 0.531 | 0.500 | 0.600 | 0.530 |
| | $He$ | 0.745 | 0.208 | 0.820 | 0.573 | 0.670 | 0.828 | 0.647 | 0.555 | 0.631 |
| | $F_{IS}$ | 0.034 | 0.106 | −0.021 | 0.056 | **0.515** | **0.362** | **0.229** | −0.082 | 0.150 |
| | $Fa$ | 0.012 | 0.042 | 0.022 | 0.019 | 0.236 | 0.176 | 0.112 | 0.045 | 0.083 |
| PV | $Na$ | 5 | 2 | 11 | 8 | 6 | 11 | 5 | 4 | 6.500 |
| | $A_E$ | 3.295 | 1.350 | 5.938 | 2.140 | 2.828 | 5.598 | 2.860 | 2.261 | 3.284 |
| | $Ho$ | 0.574 | 0.265 | 0.660 | 0.449 | 0.378 | 0.522 | 0.740 | 0.500 | 0.511 |
| | $He$ | 0.704 | 0.262 | 0.840 | 0.538 | 0.654 | 0.830 | 0.657 | 0.563 | 0.631 |
| | $F_{IS}$ | 0.186 | −0.013 | **0.216** | 0.167 | **0.425** | **0.374** | −0.128 | 0.114 | 0.168 |
| | $Fa$ | 0.087 | 0.012 | 0.104 | 0.073 | 0.190 | 0.179 | 0.074 | 0.050 | 0.096 |
| TX | $Na$ | 4 | 2 | 9 | 9 | 8 | 8 | 6 | 4 | 6.250 |
| | $A_E$ | 3.465 | 1.227 | 5.128 | 2.211 | 3.952 | 3.468 | 3.149 | 2.318 | 3.115 |
| | $Ho$ | 0.796 | 0.147 | 0.700 | 0.531 | 0.479 | 0.362 | 0.520 | 0.520 | 0.507 |
| | $He$ | 0.719 | 0.187 | 0.813 | 0.553 | 0.755 | 0.719 | 0.689 | 0.574 | 0.626 |
| | $F_{IS}$ | −0.109 | 0.218 | 0.140 | 0.041 | **0.368** | **0.500** | 0.247 | 0.095 | 0.188 |
| | $Fa$ | 0.062 | 0.079 | 0.067 | 0.043 | 0.177 | 0.242 | 0.117 | 0.047 | 0.104 |
| | $Ht$ | 0.720 | 0.217 | 0.819 | 0.552 | 0.693 | 0.796 | 0.667 | 0.559 | 0.628 |
| | $F_{IS}$ | 0.026 | 0.079 | 0.103 | 0.077 | 0.425 | 0.399 | 0.108 | 0.033 | 0.156 |

**Notes:**
Values in bold indicate significant deviations with respect to the Hardy Weinberg Equilibrium after applying the false discovery rate. Numbers below primer names are the annealing temperatures.
$Na$, Number of alleles; $A_E$, number of effective alleles; $Ho$, observed heterozygosity; $He$, expected heterozygosity; $F_{IS}$, fixation index; $Fa$, frequency of null alleles; $Ht$, total heterozygosity; C, Campeche; PV, Puerto de Veracruz; TX, Tuxpan.

vermillion snapper (*Rhombloplites aurorubens*) (*Gold, Pak & Richardson, 2001*; *Renshaw et al., 2007*).

## Genotyping

Genomic DNA was amplified by polymerase chain reactions carried out in a total volume of 10 μL, the solution contained 5.40 μL of $H_2O$, 2.0 μL of green buffer, 0.8 μL of $MgCl_2$, 0.125 μL dNTP, 0.25 μL of primer forward, 0.30 μL of primer reverse, 0.04 of Taq-polymerase and 1 μL (5 ng) of DNA extract. The Eppendorf thermocycler was programed with an initial denaturing cycle at 95 °C for 2 min, followed by 45 cycles of a denaturing step at 95 °C for 30 s, followed by an annealing step (temperature varied depending on the microsatellite; Table 1) for a period of 30 s, and a final elongation step at 72 °C for 40 s.

Allele sizes were estimated using a DNA fragment analyzer (ABI 3730xl DNA). We used ABI DS-33 dye set with G5 filter set. Forward primers were dye labeled with either 6-FAM, VIC, NED or PET dye labels; GS-600 standard set was used with LIZ dye. This protocol allowed the detection of several PCR products at the same time by

fluorescence emission. Fragment sizes were estimated using GeneMarker® software (SoftGenetics, State College, PA, USA).

## Polymorphism analysis and genetic diversity

Using the Microsatellite Toolkit software (*Park, 2008*), the Polymorphic information content (PIC) was calculated, which is an indicator of the marker quality and the degree of polymorphism in genetic cartography studies. Values of PIC higher than 0.5 indicate that the marker is highly informative, values from 0.25 to 0.5 are related to markers moderately informative and values below 0.25 indicate that the marker is slightly informative (*Botstein et al., 1980*). Genetic diversity was calculated as the number of alleles ($N_a$) and effective number of alleles ($A_E$); the latter defined as the alleles with the capacity of passing to the next generation (*Kimura & Crow, 1964*). Observed heterozygosity ($H_o$), expected heterozygosity ($H_e$) and total heterozygosity ($Ht$) were also calculated. Heterozygosity is a measure used to know the diversity of a locus and is defined as the probability that upon selecting two loci, both will be different (*Cabrero & Camacho, 2002*). These analyses were carried out with the GenAlex 6.5 software (*Peakall & Smouse, 2012*). The frequency of null alleles ($F_a$) was also estimated, considering that markers exceeding a 0.2 value should be excluded from further analyses (*Dakin & Avise, 2004*). This analysis was carried out using the MicroChecker 2.2 software (*Van Oosterhout et al., 2004*).

## Hardy-Weinberg equilibrium (HWE) and linkage disequilibrium (LD)

Wright's Fixation Index ($F_{IS}$) was used to test if the allelic frequencies conformed to the HWE (*Cockerham & Weir, 1984*). The $F_{IS}$ and $p$ values were calculated in the Arlequin 3.5 software (*Excoffier & Lischer, 2010*). LD between loci pairs was evaluated considering all individuals as a single subpopulation. LD indicates the association between loci pairs given that some alleles don't segregate in an independent manner. This was carried out using the Arlequin 3.5 software (*Excoffier & Lischer, 2010*) and the false discovery rate (*Benjamini & Yekutieli, 2001*) was applied to multiple tests of HWE and LD.

## Genetic structure

Genetic differences between subpopulation pairs were evaluated using the Wright Index ($F_{ST}$) (*Cockerham & Weir, 1984*). Given that in the present study no subpopulation had a mean of He higher than 0.9, it was not necessary to standardize $F_{ST}$, as the range of this index tends to becomes very small when He is large (*Meirmans & Hedrick, 2011*). The $R_{ST}$ index was estimated given that it is analogous to the $F_{ST}$ index, nevertheless, this index is the one that best reflects the mutation pattern of the microsatellites (*Slatkin, 1995*). This index was estimated between subpopulation pairs. The $R_{ST}$ were calculated using the Arlequin 3.5 software (*Excoffier & Lischer, 2010*).

The effective population size ($N_e$) and the number of effective migrants per generation ($N_m$) were estimated using the population parameter theta ($\theta$) and the migration parameter ($M$) was calculated between subpopulation pairs. The analyses were carried out
using the Maximum Likelihood Estimation with ten short chains and one long chain with 50,000 and 500,000 genealogies respectively at a constant mutation rate using the Migrate 3.6 software (*Beerli, 2016*).

To measure the degree of genetic among subpopulations at different hierarchic levels (between regions, among subpopulations and among individuals and within individuals) an Analysis of Molecular Variance (AMOVA) was carried out using Arlequin 3.5 software (*Excoffier & Lischer, 2010*).

### Isolation-by-distance

An association between genetic difference ($F_{ST}$) and the geographic distance can indicate restricted gene flow. We tested for this association with the Mantel test (*Aguirre-Planter, 2007*) using the GenAlex 6.5 software (*Peakall & Smouse, 2012*).

## RESULTS

### Genetic diversity

A total of 73 different alleles were recorded. The markers that resulted with the highest $N_a$ values were Prs260, Ra1, Lca107 and Prs137 (with 15, 13, 10 and 11 alleles, respectively). Based on the PIC results, six microsatellites were considered to be highly informative given that they exceed the 0.5 value, locus Prs320 (0.469) was medium level and locus Lca43 (0.239) was poorly informative (Table 2).

Mean number of effective alleles ($A_E$) varied among subpopulations, between 3.115 (*TX*) and 3.384 (*X*). Higher values were observed in the Mexican Caribbean Sea subpopulations. $H_e$ mean values for the Gulf of Mexico varied between 0.626 and 0.631 (Table 1), with the highest mean values in subpopulations *C* and PV ($H_e$ = 0.631 both). $H_e$ mean values observed in the Mexican Caribbean Sea region ranged between 0.641 and 0.655 (Table 2), with the highest mean values in subpopulations *X* and PH (0.650 and 0.655, respectively). *Ht* values were similar for the Gulf of Mexico and the Mexican Caribbean Sea, with mean values of 0.628 and 0.647 respectively.

### Genetic structure

Microsatellites that showed a significant deviation to the HWE in at least one subpopulation were Lca107, Prs137, Prs260 and Prs275 (Tables 1 and 2). In the LD results, no marker showed a significant LD after applying the false discovery rate from a total of 28 comparisons, meaning that the eight microsatellites were inherited in an independent manner. The mean frequency of null alleles for each subpopulation was between 0.045 and 0.104 (Tables 1 and 2) with no marker exceeding 0.2 of the null alleles. Given this, no marker or subpopulation was excluded from the diversity and genetic structure analyses.

Pairwise values of $F_{ST}$ ranged between 0.003 and 0.008 (Table 3), which indicates that the difference in the allelic frequencies between subpopulations are minimal. The highest values were between *CH* and both *BC* and *TX* (0.008). In subpopulation pairwise comparison, the $F_{ST}$ and $R_{ST}$ (Table 3) showed no significant difference.

**Table 2 Genetic diversity of the gray snapper (*Lutjanus griseus*) in the Mexican Caribbean Sea.**

| Sites | PIC | Lca20 0.661 | Lca43 0.239 | Prs260 0.807 | Ra1 0.561 | Lca107 0.659 | Prs137 0.774 | Prs275 0.602 | Prs328 0.469 | Mean 0.597 |
|---|---|---|---|---|---|---|---|---|---|---|
| BC | $Na$ | 4 | 3 | 10 | 11 | 7 | 8 | 5 | 3 | 6.375 |
| | $A_E$ | 3.208 | 1.446 | 5.654 | 2.407 | 3.480 | 5.013 | 2.939 | 2.224 | 3.296 |
| | $Ho$ | 0.766 | 0.326 | 0.833 | 0.612 | 0.837 | 0.776 | 0.708 | 0.510 | 0.671 |
| | $He$ | 0.696 | 0.312 | 0.832 | 0.591 | 0.720 | 0.809 | 0.667 | 0.556 | 0.648 |
| | $F_{IS}$ | −0.102 | −0.047 | −0.002 | −0.037 | −0.164 | 0.042 | −0.063 | 0.083 | −0.036 |
| | $Fa$ | 0.057 | 0.041 | 0.007 | 0.037 | 0.126 | 0.010 | 0.040 | 0.041 | 0.045 |
| CH | $Na$ | 4 | 2 | 8 | 10 | 6 | 7 | 5 | 4 | 5.750 |
| | $A_E$ | 3.476 | 1.403 | 5.020 | 2.666 | 2.807 | 4.754 | 2.944 | 2.262 | 3.167 |
| | $Ho$ | 0.766 | 0.348 | 0.813 | 0.592 | 0.548 | 0.689 | 0.708 | 0.500 | 0.620 |
| | $He$ | 0.720 | 0.290 | 0.809 | 0.631 | 0.651 | 0.799 | 0.667 | 0.564 | 0.641 |
| | $F_{IS}$ | −0.065 | −0.200 | −0.004 | 0.063 | 0.161 | 0.139 | −0.062 | 0.114 | 0.018 |
| | $Fa$ | 0.037 | 0.192 | 0.007 | 0.028 | 0.093 | 0.064 | 0.036 | 0.057 | 0.064 |
| PH | $Na$ | 4 | 5 | 11 | 9 | 7 | 8 | 5 | 4 | 6.625 |
| | $A_E$ | 3.309 | 1.491 | 6.394 | 2.900 | 3.103 | 4.469 | 2.744 | 2.328 | 3.342 |
| | $Ho$ | 0.705 | 0.370 | 0.796 | 0.580 | 0.619 | 0.600 | 0.694 | 0.571 | 0.617 |
| | $He$ | 0.706 | 0.333 | 0.852 | 0.662 | 0.686 | 0.786 | 0.642 | 0.576 | 0.655 |
| | $F_{IS}$ | 0.002 | −0.112 | 0.067 | 0.125 | 0.099 | 0.239 | −0.082 | 0.008 | 0.043 |
| | $Fa$ | 0.006 | 0.062 | 0.024 | 0.059 | 0.045 | 0.118 | 0.047 | 0.005 | 0.046 |
| X | $Na$ | 5 | 2 | 10 | 9 | 6 | 8 | 4 | 3 | 5.875 |
| | $A_E$ | 3.700 | 1.402 | 6.011 | 2.515 | 2.926 | 5.193 | 3.137 | 2.192 | 3.384 |
| | $Ho$ | 0.766 | 0.265 | 0.851 | 0.673 | 0.500 | 0.804 | 0.531 | 0.500 | 0.611 |
| | $He$ | 0.738 | 0.290 | 0.843 | 0.609 | 0.671 | 0.816 | 0.688 | 0.550 | 0.650 |
| | $F_{IS}$ | −0.039 | 0.085 | −0.010 | −0.108 | **0.259** | 0.015 | **0.231** | 0.091 | 0.066 |
| | $Fa$ | 0.029 | 0.035 | 0.013 | 0.052 | 0.121 | 0.005 | 0.107 | 0.041 | 0.050 |
| | $Ht$ | 0.712 | 0.303 | 0.833 | 0.619 | 0.679 | 0.799 | 0.666 | 0.564 | 0.647 |
| | $F_{IS}$ | −0.062 | −0.080 | 0.003 | 0.004 | 0.070 | 0.096 | −0.002 | 0.063 | 0.012 |

**Notes:**

Values in bold indicate significant deviations with respect to the Hardy Weinberg Equilibrium after applying the false discovery rate. Numbers below primer names are Polymorphic information content (PIC).

$Na$, number of alleles; $A_E$, number of effective alleles; $Ho$, observed heterozygosity; $He$, expected heterozygosity; $F_{IS}$, fixation index; $Fa$, frequency of null alleles; $Ht$, total heterozygosity; BC, Bahia de Chetumal; CH, Chiquilá; PH, Punta Herrero; X, Xahuayxol.

In the genetic flow estimation using $N_m$ values a variation from 4.1 to 25.2 was observed. High levels of connectivity were observed in all sites (Table 3).

$N_e$ values varied from 1,922 and 3,799 individuals where the highest values were observed in X and PH with 3,799 and 3,686 respectively (Table 4). Per generation migration rates ($M$) ranged between 1.3 and 11.2. Highest values were observed in the pairs CH–C (8.5), C–PV (10.0), and PV–TX (11.2) (Table 4).

The highest variation registered by AMOVA was within individuals ($PV\%$ = 98.22). The $F_{IS}$ value was 0.017, which suggests that the subpopulations present in the studied regions constitute a panmictic population. The $F_{CT}$ and $F_{SC}$ values indicated no genetic structure (Table 5).

**Table 3 Pairwise values of $F_{ST}$, $R_{ST}$ indexes between subpopulations and number of effective migrants per generations ($N_m$).**

| Spop | | $F_{ST}$ | $R_{ST}$ | | $N_m$ |
|------|------|----------|----------|------|-------|
| | | | p | SE± | |
| C | PV | 0.006 | 0.432 | 0.006 | 11.5 |
| C | TX | 0.006 | 0.648 | 0.005 | 5.6 |
| C | BC | 0.005 | 0.620 | 0.005 | 5.7 |
| C | CH | 0.007 | 0.630 | 0.005 | 16.9 |
| C | PH | 0.006 | 0.736 | 0.005 | 6.1 |
| C | X | 0.006 | 0.375 | 0.004 | 6.8 |
| PV | TX | 0.005 | 0.804 | 0.004 | 16.7 |
| PV | BC | 0.005 | 0.410 | 0.005 | 6.3 |
| PV | CH | 0.005 | 0.398 | 0.005 | 8.9 |
| PV | PH | 0.006 | 0.162 | 0.003 | 17.0 |
| PV | X | 0.003 | 0.759 | 0.004 | 7.8 |
| TX | BC | 0.006 | 0.881 | 0.003 | 4.5 |
| TX | CH | 0.008 | 0.315 | 0.004 | 5.2 |
| TX | PH | 0.007 | 0.407 | 0.005 | 4.1 |
| TX | X | 0.007 | 0.328 | 0.004 | 6.6 |
| BC | CH | 0.008 | 0.200 | 0.004 | 14.9 |
| BC | PH | 0.005 | 0.664 | 0.005 | 13.2 |
| BC | X | 0.006 | 0.121 | 0.003 | 8.1 |
| CH | PH | 0.005 | 0.333 | 0.005 | 13.3 |
| CH | X | 0.003 | 0.671 | 0.005 | 10.1 |
| PH | X | 0.005 | 0.099 | 0.003 | 25.2 |

Note:

Spop, subpopulations; p, p-value; SE, the standard error; C, Campeche; PV, Puerto de Veracruz; TX, Tuxpan; BC, Bahia de Chetumal; CH, Chiquilá; PH, Punta Herrero; X, Xahuayxol.

**Table 4 Values of the migration parameter M. In the diagonal cross section appear estimations of effective population sizes ($N_e$).**

| | C+ | PV+ | TX+ | BC+ | CH+ | PH+ | X+ |
|------|------|------|------|------|------|------|------|
| C | 2,439 | 10.0 | 2.2 | 1.9 | 8.5 | 3.2 | 1.7 |
| PV | 2.2 | 1,975 | 11.2 | 2.9 | 3.1 | 3.4 | 3.0 |
| TX | 2.8 | 6.4 | 3,058 | 1.9 | 3.0 | 1.7 | 2.8 |
| BC | 3.7 | 4.0 | 2.1 | 2,549 | 7.3 | 6.8 | 2.0 |
| CH | 11.2 | 8.4 | 1.9 | 9.7 | 1,922 | 5.2 | 3.0 |
| PH | 2.0 | 9.7 | 1.3 | 4.3 | 6.3 | 3,686 | 8.4 |
| X | 3.4 | 3.5 | 2.1 | 4.0 | 5.1 | 8.4 | 3,799 |

Note:

+, receiving subpopulation; C, Campeche; PV, Puerto de Veracruz; TX, Tuxpan; BC, Bahia de Chetumal; CH, Chiquilá; PH, Punta Herrero; X Xahuayxol.

## Isolation-by-distance

The coefficient of determination from the Mantel test was 0.0221 and the coefficient of correlations was 0.149 ($p = 0.517$), therefore, we found no relationship between the genetic and geographic distances.

**Table 5 AMOVA results.**

|                    | df  | SS        | PV%   | F-statistics        |
|--------------------|-----|-----------|-------|---------------------|
| Between regions    | 1   | 0.722     | 0.17  | $F_{CT} = 0$        |
| Among subpopulations | 5 | 8.248     | 0     | $F_{SC} = 0$        |
| Among individuals  | 341 | 560.754   | 1.94  | $F_{IS} = 0.019$    |
| Within individuals | 348 | 550.500   | 98.22 | $F_{IT} = 0.017$    |
|                    | 695 | 1,120.224 |       |                     |

**Note:**
df, degrees of freedom; SS, sum of squares; PV%, percent variance.

## DISCUSSION

The main objective of this study was to evaluate stock structure, connectivity, and to estimate genetic diversity among seven subpopulations of gray snapper (*L. griseus*) in two regions of the Mexican Atlantic: the Gulf of Mexico and the Mexican Caribbean Sea. We found no significant genetic differences between basins or among subpopulations (AMOVA: between regions and among subpopulations). Similarly, estimations of $F_{ST}$ and $R_{ST}$ between subpopulation pairs were all not significant. This lack of genetic difference indicates high genetic connectivity among all subpopulations of *L. griseus* along this ~1,950 km stretch of coastline. This result is also supported by the Mantel Test ($p > 0.05$), which didn't show an association between genetic differences and geographic distances. In all cases, the number of migrants per generation ($N_m$) entering any given subpopulation was greater than four, this indicates that there is unrestricted gene flow and that the populations behave like a panmictic population, as theoretically with the migration of a single organism ($N_m = 1$) allele fixation is avoided (*Slatkin, 1995*). However, our estimates of *M* appear to be greater within basins than between basins, and in the Gulf of Mexico region there appears to be a distinct directionality to migration, with greater *M*-values between neighboring subpopulation pairs going west compared to east (Table 4; Table S1). This suggests that despite the fact that $F_{ST}$ based estimates of gene flow show no regionalization, there may be some subtle patterns of reduced connectivity between basins and perhaps a distinct directionality of gene flow in the Gulf of Mexico.

The fact that we found that connectivity may be slightly restricted between the Mexican Caribbean and southern Gulf of Mexico is aligned with findings from previous studies of connectivity between the regions. Blacktip sharks (*C. limbatus*) are known to show strong genetic differentiation between the Mesoamerican Barrier Reef System (Caribbean) and the southern Gulf of Mexico (*Keeney et al., 2005*). Additionally, bicolor damselfish (*S. partitus*) also shows evidence of a weak restriction in gene flow between the Mexican Caribbean and southern Gulf of Mexico (*Villegas Sánchez et al., 2014*). Our finding here of generally greater migration rates among subpopulations within regions than across regions further supports these studies.

Ocean currents and the biology of *L. griseus* may play an important role in connectivity among the subpopulations. A particle tracking model of a closely related species *Lutjanus analis* virtual larvae carried out by *Martínez, Carrillo & Marinone (2019)* suggests that the marine protected areas of the Mesoamerican Reef network are all highly connected

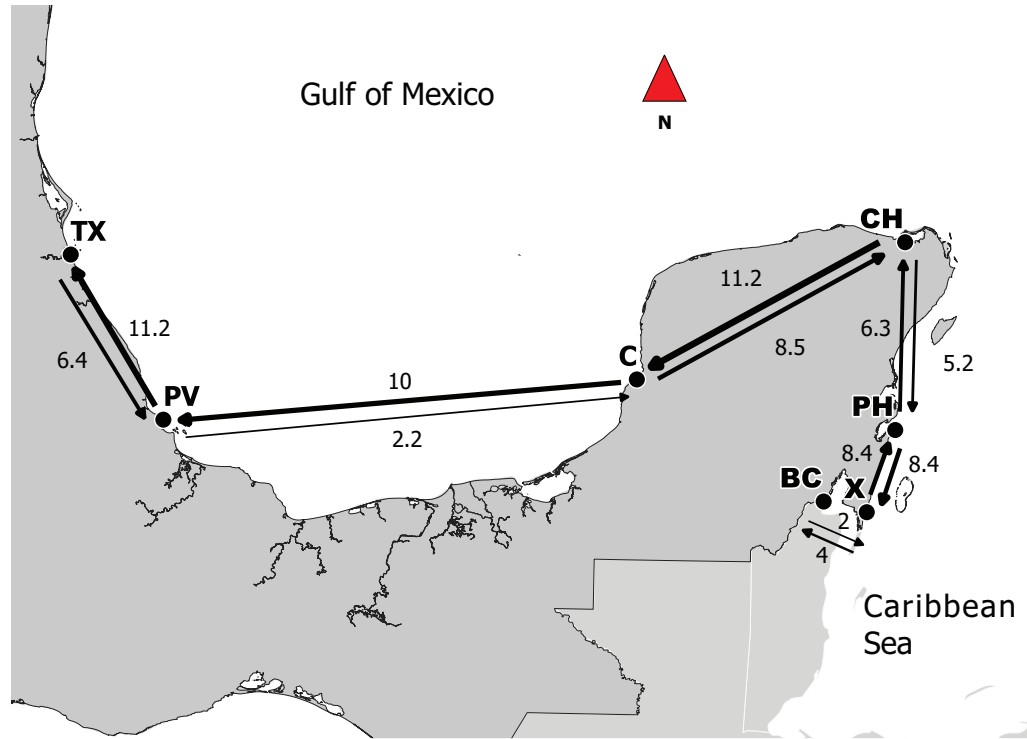

**Figure 2 Directionality of migration in the Caribbean and the Gulf of Mexico between adjacent sites.** Based on the results obtained from Migrate, lines represent migration between points; thicker lines represent stronger levels of genetic flow and the values correspond to MLE. Abbreviations: Campeche (*C*), Puerto de Veracruz (*PV*), Tuxpan (*TX*), Bahia de Chetumal (*BC*), Chiquilá (*CH*), Punta Herrero (*PH*) and Xahuayxol (*X*).

through ocean current. This species has a similar life cycle as *L. griseus*, therefore the results of this project support the findings by these authors. The Yucatan current (average speed of 1.5 m/s), which later becomes the Loop Current (*Athié et al., 2011*), can export fish larvae from the Caribbean up to the Gulf of Mexico (*Carrillo et al., 2015*). Possibly these ocean dynamics favor the dispersion of *L. griseus* larvae. In adults, migrations are present but these are mainly local, with registered distances between 35 and 122 km (*Claro & Lindeman, 2008*), and these movements are typically between inshore habitats and shelf habitats rather than among reefs. So it is unlikely that adult movements alone are resulting in high levels of gene flow observed in this study.

In terms of the apparent directionality of gene flow in the southern Gulf of Mexico, studies of oceanographic connectivity in the southern Gulf of Mexico indicate that east-gene flow from Campeche Bank to Veracruz and Tuxpan reefs is low and that connectivity is stronger going west to east (*Sanvicente-Añorve et al., 2014*). The pattern of migration observed in this study appears to contradict these previous findings (Fig. 2). However, ocean currents in this region are complex with the presence of eddies and a seasonal shifts in direction within the inner shelf with summer months showing flux from east to west and winter months showing flux from west to east (*Salas-Monreal et al., 2017*). The summer period of greater east to west connectivity also coincides generally

with gray snapper spawning season (*Domeier, Koenig & Coleman, 1996*). The pattern of east-to-west migration observed here is a feasible explanation.

The highest migration rates were between *CH–C* and *C–PV*. According to this result, Campeche can be an important point for the connectivity between the two regions. It is important to mention that this Mexican state has the largest extent of mangroves (*Lara-Lara et al., 2008*), and it has been shown that mangroves are the principal habitats for larvae and juveniles of *L. griseus*, even though they also use seagrass beds at times (*Claro & Lindeman, 2008*). For successful sea dispersion, it is essential that larvae find an adequate habitat for recruitment (*Cowen, 2006*). Thus, if the habitat is fragmented or destroyed, the connectivity can be limited (*Jones, Srinivasan & Almany, 2007*). We also found high migration rates between subpopulations that were apparently counter to the flow of main currents (e.g., *C–CH, CH–BC* and *PH–X*). This could be due to displacing reproductive aggregations that occur between the months of June and August around the new moon on the shelf border with a duration period of 8–10 days (*Claro & Lindeman, 2008*). Moreover, it has been reported periods of a coastal countercurrent over the shelf (*Carrillo et al., 2017*) that could promote migrations as it was found in our results.

The effective population sizes estimated from Migrate (θ) were large, however, they must be considered with some caution, especially when interpreting for management and conservation issues. The mutation rate that is, considered for the $N_e$ calculation can create a bias with different values with several orders of magnitude. It is suggested that the values of $N_e$ should only be taken as comparative ones between sampled sites rather than a true point estimate (*Beerli, 2016*).

Genetic diversity in these populations was high (*Ht* general average = 0.640; PIC general average = 0.597). According to the PIC, the microsatellites used are highly informative and polymorphic (*Botstein et al., 1980*). The number of alleles per microsatellite varied between 5 (Lca43) and 15 (Prs260), with a general mean of 6.17 ($n = 8$); previous studies of *L. griseus* have reported means of 5.43 ($n = 14$) (*Renshaw et al., 2007*) and 7.0 ($n = 14$) (*Gold et al., 2009*), similar to the values reported in this study. These subpopulations appear to be similar in diversity to those of the U.S. Gulf of Mexico and southern Florida.

In this study, deviations to the HWE were observed for several loci in some subpopulations with mean values of $F_{IS}$ varying from 0.066 (*X*) to 0.188 (*TX*), which suggests a heterozygote deficit not previously reported for *L. griseus*. MicroChecker indicated that the most possible cause for the significant values in the $F_{IS}$ index could be the presence of null alleles. Null alleles are produced when there is a mutation in one of the primer binding sites of the microsatellites; this prevents annealing with the designed primer preventing the amplification of one of the alleles, resulting in a false homozygote (*Estoup et al., 1998*; *Chapuis & Estoup, 2007*). Such condition is frequent in microsatellites given their high levels of polymorphism. This has been reported in various species especially in populations with high effective size (*Neff & Gross, 2001*; *Chapuis & Estoup, 2007*). Additionally, fish have a high mutation rate in comparison with other classes (reptiles, birds, amphibians and mammals), because they have larger microsatellites and length is an important factor that influences mutation rate (*Neff & Gross, 2001*). There are ecological explanations for deviations from HWE including inbreeding and

genetic drift. The deficiency in this case is unlikely to be explained by inbreeding due to the reproductive habits of *L. griseus*, given that they form aggregations and in these, gametes are simultaneously released to be fertilized (*Claro & Lindeman, 2008*). However, heterozygote excesses have been explained by sweepstakes recruitment in marine organisms with small pelagic larvae period and highly variable adult reproductive success (*Hedgecock, 1994*) which can lead to instantaneous genetic drift.

## CONCLUSIONS

The high levels of genetic diversity, similar to those observed in other gray snapper populations from the northern Gulf of Mexico, as well as the high levels of gene flow in general, suggest that *L. griseus* constitutes a single genetic population in the Mexican Caribbean and the southern Gulf of Mexico. The absence of population bottlenecks or disturbances on its connectivity across this large region should be taken into account to define a proper management for the stock. More work is needed to verify the connectivity and the apparent unidirectionality in gene flow in the southern Gulf of Mexico (east-to-west). The next priority for understanding gray snapper populations in the Western Atlantic is to determine the degree of connectivity between Mexican populations and the northern Gulf of Mexico and between Mexico and Cuba.

## ACKNOWLEDGEMENTS

We thank Mauricio Iván Espadas Alcocer and Roberto Zamora Bustillos for providing laboratory and field assistance with the samples obtained in the Mexican Caribbean Sea. Jason Selwyn for his support in the laboratory at TAMUCC. Chloe Rosas and Dariel Correa for the translation of the document.

### Funding

This work was supported by the Tecnológico Nacional de México (Project Number: 6415.18-P) and a CONACYT scholarship with number 636312. The funders had no role in study design, data collection and analysis, decision to publish, or preparation of the manuscript.

### Grant Disclosures

The following grant information was disclosed by the authors:
Tecnológico Nacional de México: 6415.18-P.
CONACYT scholarship: 636312.

### Competing Interests

The authors declare that they have no competing interests.

### Author Contributions

- Oscar de Jesús Rosado-Nic conceived and designed the experiments, performed the experiments, analyzed the data, prepared figures and/or tables, authored or reviewed drafts of the paper, and approved the final draft.

- J. Derek Hogan conceived and designed the experiments, performed the experiments, analyzed the data, prepared figures and/or tables, authored or reviewed drafts of the paper, and approved the final draft.
- José Héctor Lara-Arenas analyzed the data, prepared figures and/or tables, authored or reviewed drafts of the paper, and approved the final draft.
- Rigoberto Rosas-Luis analyzed the data, authored or reviewed drafts of the paper, and approved the final draft.
- Laura Carrillo analyzed the data, authored or reviewed drafts of the paper, and approved the final draft.
- Carmen Amelia Villegas-Sánchez conceived and designed the experiments, performed the experiments, analyzed the data, prepared figures and/or tables, authored or reviewed drafts of the paper, and approved the final draft.

## Field Study Permissions

The following information was supplied relating to field study approvals (i.e., approving body and any reference numbers):

Field experiments were approved by the SEMARNAT (23/K4-0002/05/18).

## Data Availability

Raw data measurements are available in a Supplemental File.

## Supplemental Information

Supplemental information for this article can be found online at http://dx.doi.org/10.7717/peerj.8485#supplemental-information.

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
