# Peer review of "Gene flow between subpopulations of gray snapper (Lutjanus griseus) from the Caribbean and Gulf of Mexico"

_PeerJ, doi:10.7717/peerj.8485_

## Round 0.1 · original submission · Minor Revisions

Two expert reviewers have evaluated your manuscript and their comments can be seen below and in an attached PDF. Both have favourable comments about the manuscript, as well as some suggestions for improvement in a revised version.

Reviewer 1 ·

Basic reporting

It's an interesting manuscript, using standard methods on a different place, with a species without overfishing nor protection status. I suggest considering two methodological observations: 1) a common mistake in population genetic studies is to violate the independence assumption and 2) considering that authors didn't identify genetic subpopulations, analyze the potential presence of barriers through appropriate statistical tests.

Experimental design

No comment

Validity of the findings

Authors must discuss the effect of the lack of deviations to HWE in some microsatellites used in their analysis.

Additional comments

It's an interesting work that will help to explain the connectivity among the Gulf of Mexico and the Caribbean, but need to be reinforced in some areas.

Annotated reviews are not available for download in order to protect the identity of reviewers who chose to remain anonymous.

Reviewer 2 ·

Basic reporting

The authors have conducted a meticulous study on the population genetic structure of gray snapper (Lutjanus griseus) from the Caribbean and Gulf of Mexico.
The major finding is the high levels of genetic diversity and high rate of migration suggest that these populations of gray snapper in the Mexican Caribbean and southern Gulf of Mexico may be fairly robust to fishing pressure. Furthermore, gene flow patterns found in this study are certainly a step forward in our knowledge of the population dynamics of that species which is useful for establishing appropriate management and conservation.
The manuscript was well written, with informative figures and tables and thorough results. The language used is clear. The literature is well referenced and the conclusions based on the results are generally sound, however, citing some studies of 2016 to date in the introduction and discussion sections would be beneficial and enhance the manuscript.
In some instances, some changes should be taken into account to add consistency to the text:
1) Line 73-75. It is missing a reference for this statement.
2) Line 254. Replace “insignificant” with “not significant”.
3) Line 277. I suggest to replace word “large” with “important” to be clear.
4) Line 298. Eliminate “at least” do to ambiguity.
5) Line 331. Replace the word “fishes” with “fish”.
6) Figure 2. The header abbreviations should not be in italics. That means It be should be consistent with the figure abbreviations. Furthermore, I suggest either an abbreviation font size (into the figure) to be larger so it is easily visualized.
7) Table 1. The “Gulf of Mexico” label could be placed on the top of the table, otherwise, it is confusing where it is. Please consider an alternative column header.
8) Table 2. The “Mexican Caribbean Sea” label could be placed on the top of the table, otherwise, it is confusing where it is. Please consider an alternative column header.

Experimental design

The study is pertinent to know the diversity and genetic structure of populations of L. griseus that is a commercially important species. That allows us to advance our knowledge of the vulnerability of this species to the fishing, which is useful for establishing appropriate management and conservation of stocks.
Sample sizes and analyses used were generally adequate to reach the objectives, however, some suggestions should be taken into account for the robustness of the study:
1) Lines 141-143. Please provide more information to be clear what do you mean by “capture resident individuals and how did avoid overlapping with the migration activities of these organisms”, or make an explicit statement about migration activities of this species in order to understand the sampling design background.
2) I suggestion computing a PCA using genetic frequencies in order to identify the most divergent sites (if there would be), since multivariate methods summarize the genetic variability without making strong assumptions about an evolution model, that means they do not rely on Hardy–Weinberg equilibrium, nor do they suppose the absence of linkage disequilibrium. Furthermore, it would benefit by analyzing the genetic structure using a Bayesian approach with STRUCTURE software, although you already didn't find genetic differentiation you should to explore your data to get more information about the population structure since you found genetic flow patterns.
3) It would improve the quality of the manuscript the inclusion of a standardized measure of genetic differentiation (F'ST) because genetic diversity associated with microsatellites can affect FST values. See:
Meirmans, P. G., & Van Tienderen, P. H. (2004). GENOTYPE and GENODIVE: two programs for the analysis of genetic diversity of asexual organisms. Molecular Ecology Notes, 4(4), 792-794.
Meirmans, P. G. (2006). Using the AMOVA framework to estimate a standardized genetic differentiation measure. Evolution, 60(11), 2399-2402.
Meirmans, P. G., & Hedrick, P. W. (2011). Assessing population structure: FST and related measures. Molecular ecology resources, 11(1), 5-18.

Validity of the findings

The findings were supported by the genetic analyses that were performed, however, it would be very important to carry out other statistical methods to robust the study. Line 260. Yours findings suggest a panmictic population, they don’t show a metapopulation structure. “Metapopulations differ from the other structures in having regular extinction and recolonization events, source and sink populations. Metapopulations have different genetic consequences, typically having the worst genetic impacts (inbreeding and loss of genetic diversity) for the same total population sizes, especially in the long term”. See:
Crooks, K., & Sanjayan, M. (2006). Connectivity Conservation. (pp. 712). United
States of America, New York, Cambridge Univesity Press.
1) Lines 261-267, 349-350. Both "Some subtle patterns of reduced connectivity between basins and a distinct directionality of gene flow in the Gulf of Mexico" and "east-west gene flow would indicate that Tuxpan reefs maybe a sink population", are a signal of a metapopulation structure, nevertheless, you should be cautious to the interpretation of results.
2) Line 316. Please, add more information about the “polymorphic information content general average (0.597)”, you didn't mention it in the methods neither made reference to any table.
3) Line 323. Both table 1and table 2 are not showing values average for the FIS index higher than 0.20, in that case, you should rewriter the discussion regarding real values the FIS index.
4) Lines 345-348. Although high levels of connectivity found across the Mexican Caribbean and the southern Gulf of Mexico suggest that the genetic population of L. griseus is quite large and may be fairly robust to fishing pressure, the conclusion could be improved with a more detailed discussion of the background and a more cautious interpretation of results. Currently, the manuscript does not adequately address this issue.

Additional comments

I commend the authors for a meticulous study on the population genetic structure of gray snapper (Lutjanus griseus) from the Caribbean and Gulf of Mexico. The findings of this study are certainly a step forward in our knowledge of the population dynamics of that species commercially important, which is useful for establishing appropriate management and conservation. The conclusions based on the results are generally sound.

---

## Round 0.2 · Minor Revisions

I am satisfied with the changes made to the manuscript. However, there is one commnet from Reviewer 1 that was not attended to. With reference to the sentence that ranges from lines 93 to 99 in the track changes version you did not change the reference or answer the comment : "Neither the cited work nor this make studies to give their opinion about the overfishing and the effect of lack of regulations". Please make sure to do so.

---

## Round 0.3 · accepted · Accept

I am satisfied with the references that were added.